DOI: 10.1038/s41467-018-04394-3　　**OPEN**

Corrected: Author correction

# Operando monitoring the lithium spatial distribution of lithium metal anodes

Shasha Lv[1,2], Tomas Verhallen[1], Alexandros Vasileiadis[1], Frans Ooms[1], Yaolin Xu [1], Zhaolong Li[1], Zhengcao Li[2,3] & Marnix Wagemaker[1]

Electrical mobility demands an increase of battery energy density beyond current lithium-ion technology. A crucial bottleneck is the development of safe and reversible lithium-metal anodes, which is challenged by short circuits caused by lithium-metal dendrites and a short cycle life owing to the reactivity with electrolytes. The evolution of the lithium-metal-film morphology is relatively poorly understood because it is difficult to monitor lithium, in particular during battery operation. Here we employ operando neutron depth profiling as a noninvasive and versatile technique, complementary to microscopic techniques, providing the spatial distribution/density of lithium during plating and stripping. The evolution of the lithium-metal-density-profile is shown to depend on the current density, electrolyte composition and cycling history, and allows monitoring the amount and distribution of inactive lithium over cycling. A small amount of reversible lithium uptake in the copper current collector during plating and stripping is revealed, providing insights towards improved lithium-metal anodes.

[1] Department of Radiation Science and Technology, Delft University of Technology, Mekelweg 15, Delft 2629 JB, The Netherlands. [2] The State Key Laboratory of New Ceramics and Fine Processing, School of Materials Science and Engineering, Tsinghua University, Beijing 100084, China. [3] Key Laboratory of Advanced Materials (MOE), School of Materials Science and Engineering, Tsinghua University, Beijing 100084, China. Correspondence and requests for materials should be addressed to Z.L. (email: zcli@tsinghua.edu.cn) or to M.W. (email: m.wagemaker@tudelft.nl)

Based on their large gravimetric and volumetric energy densities, Li-ion batteries are the technology of choice for portable electronics and electrical mobility[1–3]. The positive and negative electrodes in Li-ion batteries are able to store Li, the specific weight of which is a decisive factor in the energy density. As the energy density of Li-ion insertion chemistries is approaching its limit, intensive research is directed toward high-capacity anodes and cathodes. The cathode alternatives that are intensively studied include the Li-S and Li-$O_2$ conversion chemistries[4–7], whereas the ultimate anode is Li-metal having the highest specific capacity for Li (3860 mAh g$^{-1}$), > 10 times larger than standard graphite anodes (370 mAh g$^{-1}$), and the lowest redox potential (−3.04 V vs. standard hydrogen). For this reason, Li-metal was already investigated intensively in the early stages of Li battery research[8,9]. However, the safety risks associated with dendrite formation and the success of graphite as anode largely took away the focus from Li-metal anodes. The major challenges for Li-metal anodes are safety and cycleability, directly related to its tendency to be deposited in a dendritic and mossy form and its high reactivity towards common electrolytes[2,8,10,11]. When dendrites penetrate the separator/electrolyte and reach the cathode, this causes an internal short-circuit that may induce rapid spontaneous discharge and consequential safety hazards. Recently, a diversity of promising strategies has been developed and proposed, either aiming a preventing, suppressing, or blocking dendrite formation[11–14]. These strategies are mainly based on our current understanding of electrochemical Li-metal plating, which is now rapidly increasing through fundamental studies, in particular employing microscopic and optical techniques[11–14]. Pei et al.[15] demonstrated with scanning electron microscopy (SEM) that the Li-metal nuclei density is proportional to the cubic power of the overpotential, consistent with classical nucleation and growth theory. Cryogenic TEM revealed the preferred growth facets of Li-metal and the nature of the solid electrolyte interface (SEI) nanostructures in various electrolytes[16]. SEM and optical studies have shown a correlation between the current density and the plated Li-metal microstructure in line with the Chazalviel model[17]. This model predicts that at current densities that deplete the electrolyte concentration, at the surface of the anode, the anion concentration will eventually drop to zero after a characteristic Sand's time time[18]. Under these conditions, plating becomes inhomogeneous and self-amplified growth of dendrites is induced[17–19]. Below the critical current, Li deposits dominantly as whiskers in carbonaceous electrolytes[18], which is believed to be the result of the formation of the SEI on the surface, resulting in porous Li-metal/SEI heterogeneous morphologies. However, the situation appears very complex as dendrites also appear to grow far below the critical current[11–14,20,21]. The microstructure evolution during battery operation determines the exposure rate of fresh Li-metal surface to the electrolyte. The induced SEI formation through electrolyte reduction lowers the coulombic efficiency and raises the internal resistance, causing reduced cycling capacities and early cell death. In addition, the microstructural evolution during stripping may leave isolated regions of Li-metal. This so called "dead" Li-metal contributes to capacity loss, and may have an important role in the penetration of Li-metal though the electrolyte towards the positive electrode over repeated cycling, causing an internal short-circuit and the consequential safety risks. Therefore, the evolution of the complex SEI/Li-metal microstructure should be considered to assess the safety risks and energy storage efficiency of Li-metal batteries.

Developing understanding of the Li anode microstructures is challenged by the difficulty to detect Li, both its quantitative distribution and chemical form, in particular during realistic battery operation[11,14–25]. This has motivated the use and development of several microscopic and spectroscopic characterization approaches, mostly under ex situ or in situ conditions[11,12,26] as operando characterization is even more challenging. In situ TEM microscopic studies have been able to observe the local plating reactions, including local dendrite growth and SEI formation[27–33]. Typically, these in situ open cell TEM as well as SEM[34,35] experiments are limited to low volatile electrolytes (e.g., ionic liquids and solid electrolytes). Recently, nanoscale imaging was achieved with in situ TEM for liquid cells[36], which allowed detailed local characterization of the Li-metal plating[37] and SEI phases[38,39], although care should be taken for the potential influence of the electron beam on the electrochemical reactions[40,41]. In situ and operando optical microscopy has been used to observe the evolution of dendrites[18,19,42,43], as well as using laser scanning confocal microscopy[44,45], giving direct insight in the factors that influence dendritic deposition of lithium. Operando $^7$Li NMR spectroscopy has been shown to be a promising approache to monitor the nature of the Li species[46], whereas in situ $^7$Li NMR imaging was shown to be able to measure the Li-metal microstructure buildup with sub-micron resolution[47]. Synchrotron hard X-ray microtomography experiments were able to observe Li dendritic structures during the early stage of dendrite formation polymer cells[48,49]. Despite these crucial advances in situ and operando characterizations, Li-metal research would benefit from the development of quantitative and non invasive techniques that operate under realistic operando battery conditions.

Here we apply Neutron Depth Profiling (NDP)[50,51], complementary to microscopic and spectroscopic techniques, providing quantitative, noninvasive operando measurement of the Li-ion density as a function of depth in Li-metal anodes. The plating and stripping activity in liquid electrolyte symmetric cells, utilizing copper foil current collectors, is asymmetric, where during plating Li-metal extends progressively into the electrolyte. In contrast, Li-metal stripping occurs more homogeneously, which can be held responsible for the formation of isolated regions of inactive or "dead" Li-metal, lowering the Coulombic efficiency. The initially forming Li-metal morphology, the density of which strongly depends on the current density, templates the concurrently forming SEI. As a consequence, the morphology of the SEI strongly depends on the initial current rate, which influences the Li-metal morphology as well as the reversible Coulombic and Li efficiency on subsequent cycles. Unexpectedly, a small amount of reversible Li insertion and extraction is observed in the copper current collector, providing new insights in current collector aging and how to prevent this relatively unknown phenomena.

## Results

**Principle of operando NDP Li-metal.** NDP of Li relies on the capture reaction of a thermal neutron with a $^6$Li atom, which results in two charged particles that, based on conservation of momentum and energy, have a well-defined energy: $^4$He (2044 keV) and $^3$H (2727 keV). By placing a detector at some distance from a Li-containing battery as shown in Fig. 1a, the energy loss of these charged particles can be related to the depth of the capture reaction. Hence, the original position of the $^6$Li can be determined through the stopping power of the material. Because each measured $^4$He and $^3$H particle represents one Li in the system of interest, NDP measures the Li number density with great accuracy as a function of depth. Based on this ability NDP has been used to monitor Li distributions in thin film solid state micro-batteries[52], in Sn  and Al electrodes[51,53], in LiFePO$_4$ electrodes[54,55], and Li-metal plating at the interface of a garnet solid electrolyte[56].

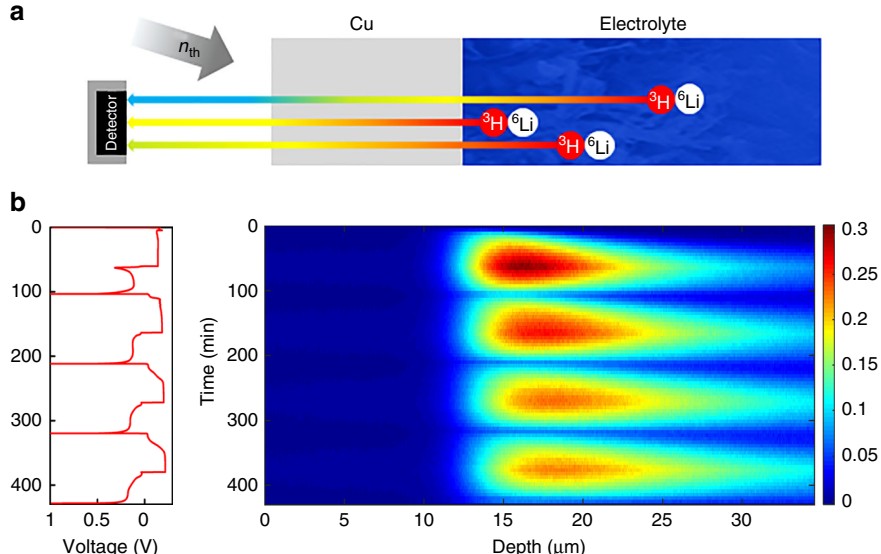

**Fig. 1** Principle of operando neutron depth profiling of Li-metal plating and stripping. **a** principle and schematic setup of operando neutron depth profiling (NDP). **b** Operando NDP measurements of four plating and stripping cycles at 1.0 mA cm$^{-2}$ current density, showing the fractional Li density as a function of depth perpendicular to the Cu current collector. The fractional Li density is obtained by normalizing the measured Li density by the Li-metal density

In the present research, operando NDP is used to investigate Li plating and stripping on a Cu current collector in symmetric Li-metal cells employing a LiPF$_6$ ethylene carbonatedimethyl carbonate (EC/DMC) liquid electrolyte. Because natural Li consists of only 7.5% of the $^6$Li isotope, enriched Li-metal was used (95% $^6$Li and 5% $^7$Li), which is assumed not to influence the metal and plating reactions. By bringing the electrolyte in contact with excess of enriched Li-metal, also the electrolyte is enriched up to 95% $^6$Li and 5% $^7$Li. Because the $^4$He particles are not able to pass the Cu current collector, only the $^3$H particles are detected during the present experiments. The depth as shown in Fig. 1b is measured starting from the interface of the Cu current collector with the ambient atmosphere. The interface of the Cu current collector with the interior of the cell is observed by the appearance of Li, and is positioned at ~ 12 μm, reflecting the Cu current collector thickness. The 12 μm thick Cu foil results in a loss of 1040 keV of the initial 2727 keV of the $^3$H. In Fig. 1b operando NDP is shown for four subsequent electrochemical plating and stripping cycles at 1 mA cm$^{-2}$ up to a plating capacity of 1 mAh cm$^{-2}$ during which continuously 1 min NDP measurements are acquired. During plating the detected Li density on the 12 μm thick Cu current collector reflects the anticipated appearance of Li, whereas the Li density decreases during the stripping current as anticipated. The background and intensity correction as described in the methods sections result in the quantitative measurement of the Li concentration as a function of depth. Figure 1b shows the Li density normalized to Li-metal; hence, it reflects the fractional density with respect to Li-metal. The maximum fractional density is ~ 0.3, which, assuming this all to be Li-metal, results in a minimum porosity of the deposited Li of ~ 0.7 (70%). This demonstrates that plating results in a highly porous Li-metal film, consistent with the plated capacity, 1 mAh cm$^{-2}$, which corresponds to a solid Li-metal film thickness of ~ 5 μm, in this case distributed over at least a 20 μm thick layer as observed in Fig. 1b. From the evolution of the Li density, the growth rate can be determined, amounting 0.23 μm min$^{-1}$ for 0.1 fractional density and 0.11 μm min$^{-1}$ for 0.2 fractional density. Very little quantitative operando information exists on the Li density and porosity of electrochemically plated Li, illustrating the added value of NDP for the characterization of Li-metal-

plating reactions. The depth resolution for these systems is ~ 70 nm, which is dictated by the stopping power of the materials along the path of the $^3$H between the Li position and the detector. Therefore, NDP results in the Li density, and approximate porosity of the deposited Li-metal with a relatively good spatial depth resolution.

**Impact of current density and salt concentration**. The Li density profiles in Fig. 1b show that repeated cycling leads to less-dense, and thicker Li deposits. The low-density tails of these profiles indicate highly porous Li morphologies that extend far into the electrolyte. Because Li-metal reduces the EC/DMC electrolyte[57,58], resulting in a heterogeneous mixture of inorganic compounds (Li$_2$O and Li$_2$CO$_3$ for carbonate electrolytes) and organic polymers[16,59], the observed Li density is the sum of Li-metal and Li in the SEI at the surface of the Li-metal. NDP is unable to distinguish the chemical nature of the observed Li. For comparison with the NDP experiments, ex situ SEM measurements were performed that show the micro structured morphologies of the deposited Li-metal in Fig. 2a–h at different deposition capacities. Similar to other investigations, at 0.1 mAh cm$^{-2}$ and at 0.2 mAh cm$^{-2}$ Li deposit "hotspots" appear. At a capacity of 0.5 mAh cm$^{-2}$ elongated Li deposits as well as mossy Li structures are formed and at 1.0 mAh cm$^{-2}$ ~ 200 nm wide needle-like tips are observed. Figure 2f shows that after stripping, a porous microstructure remains. The tilted SEM image at 1 mAh cm$^{-2}$, Fig. 2g, clearly shows a porous morphology including dendrites extending ~ 13 μm from the Cu current collector into the electrolyte. This is in good agreement with the average thickness of the Li density observed with operando NDP, where the first plating and stripping cycle at mAh cm$^{-2}$ is displayed in Fig. 2i. The dense Li region observed by NDP in Fig. 1b corresponds to the more dense mossy Li deposits observed by SEM, whereas the low-density tails extending far into the electrolyte corresponds to the dendrites observed by SEM. After stripping, the SEM shown in Fig. 2f displays difficult to characterize morphologies, most likely representing a mixture of SEI species and "dead" Li-metal. The evolution of the inactive Li, both "dead" Li-metal and Li in the SEI during cycling as observed by NDP is discussed in more detail in the next section.

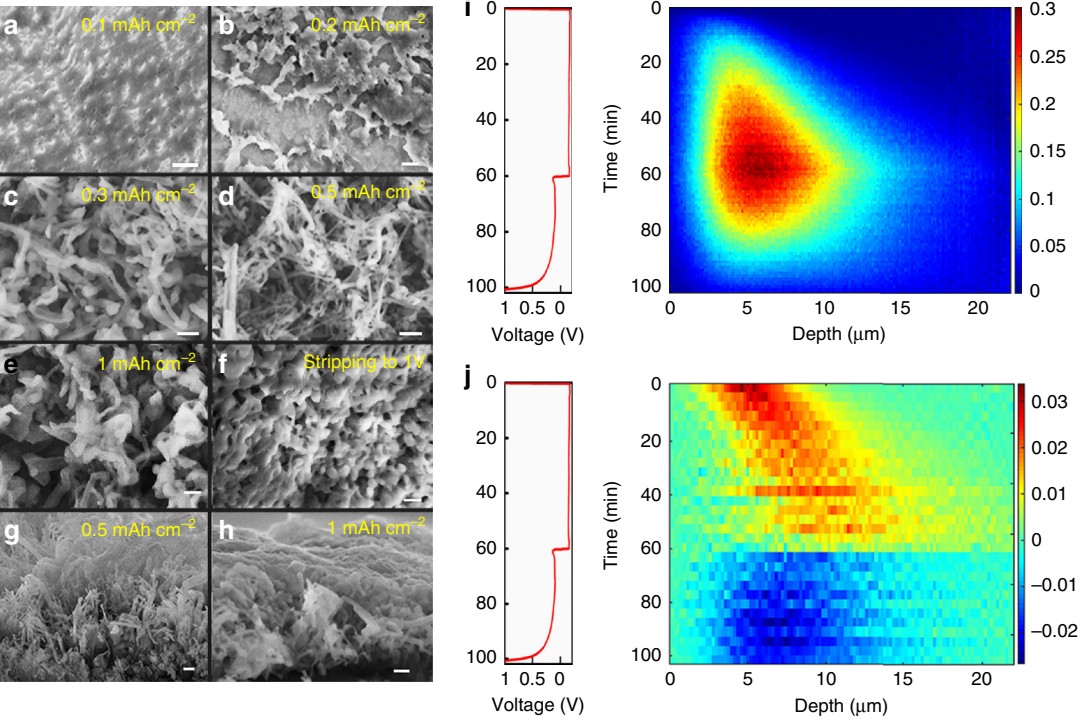

**Fig. 2** Li density and Li-metal-plating/stripping activity by operando NDP compared with ex situ SEM. **a–h** SEM measurement showing the 1 mA cm$^{-2}$ Li-plated metal morphology at **a** 0.1, **b** 0.2, **c** 0.3, **d** 0.5, and **e** 1 mAh cm$^{-2}$ and **f** after stripping. **g, h** Cross sections are shown for 0.5 and 1 mAh cm$^{-2}$. The scale bar in **a–b** is 5 μm and in **c–h** it is 1 μm. **i** Displays the Li density vs. time from operando NDP during the first plating and stripping cycle at 1 mAh cm$^{-2}$ and **j** the Li-plating and stripping activity, which is derived from the change in Li density upon each time step in **i**. In **i** and **j** the depth is measured starting from the interface of the Cu current collector with the electrolyte/SEI/Li-metal

From the 2D Li density in Fig. 2i, the plating and stripping process appears to be asymmetric. This can be more clearly visualized by subtracting subsequent Li densities in time, resulting in the change in Li density shown in Fig. 2j. This provides direct insight in the distribution of the plating and stripping, resulting in a positive change in Li density during plating (yellow/red), and a negative change in Li density during stripping (blue). During plating at 1 mA cm$^{-2}$ Li deposition is localized within 5 μm in depth, the position of which progressively moves away from the Cu current collector into the electrolyte. This is consistent with root growth of mossy Li pushing the moss progressively into the electrolyte, as observed by optical analysis[11–14,18]. The localized growth indicates that at these conditions the root growth dominates, as compared with the thickening of the mossy whiskers, because thickening would result in a homogeneous Li-plating activity. In contrast, the stripping activity is more homogeneously distributed over the thickness of the deposited Li-metal film. This indicates thinning of the mossy Li whiskers throughout the deposited Li-metal film. Thinning is likely to result in regions of Li-metal that are disconnected from the Cu current collector, so called "dead" Lithium, which lowers the capacity upon cycling. Thereby, the average plating and stripping activity observed by operando NDP provides a very direct and operando view on the average growth and stripping mechanism, indicating that the fundamental origin of "dead" Li is the homogeneous stripping activity throughout the depth.

The local ion concentration is considered to be one of the key parameters that determine the growth mechanism and Li-metal morphology under plating conditions. As a consequence, the Li-metal morphology will depend on the current density. At current densities that lead to ion depletion at the metal surface, inhomogeneous Li-metal deposition in the form of dendrite formation is induced[17–19]. Although this prediction appears to

works well above the critical current, dendrites are also observed at much lower current densities such as studied at present[11–14], and also observed in Fig. 2a–h. How the Li-metal mossy/dendrite density distribution depends on the current density and the Li salt concentration can be directly assessed by operando NDP. More dendritic Li-metal growth may be expected to result in less-dense Li-metal films extending further away from the current collector as suggested by optical studies[11–14].

In Fig. 3, operando NDP is shown during the first plating/stripping cycle at both 0.5 and 2 mA cm$^{-2}$ for 2 and 0.5 h, respectively, (both resulting in 1 mAh cm$^{-2}$ plated capacity). This reveals that a larger current density results in more compact Li-metal plating, which is difficult to assess by comparing the SEM images in Fig. 2g, h. For mossy/dendritic features this result is hard to validate with for instance SEM or optical techniques, and at first appears to contradict that larger currents are expected to result in more dendritic growth, and hence less-dense Li-metal films. The present operando NDP indicates that there is a strong relationship between the current and the resulting Li-metal density, even at these relatively low current densities. The plating and stripping activity shown in Fig. 3 demonstrates that plating occurs much more localized and closer to the current collector at 2 mA cm$^{-2}$ compared with 0.5 mA cm$^{-2}$ current density. This may be explained by current density dependent Li-metal nucleation behavior. Pei et al.[15] have shown that the Li nuclei size in ether electrolytes is inversely proportional to the overpotential and that the number of Li nuclei is proportional to the cubic power of the overpotential, following classic nucleation theory. The much higher overpotentials at 2 mA cm$^{-2}$, see Fig. 3, can therefore expected to result in denser plating at larger current densities based on a much larger amount of Li-metal growth centers, in particular in the vicinity of the Cu current collector. Another aspect is the SEI, formation of which

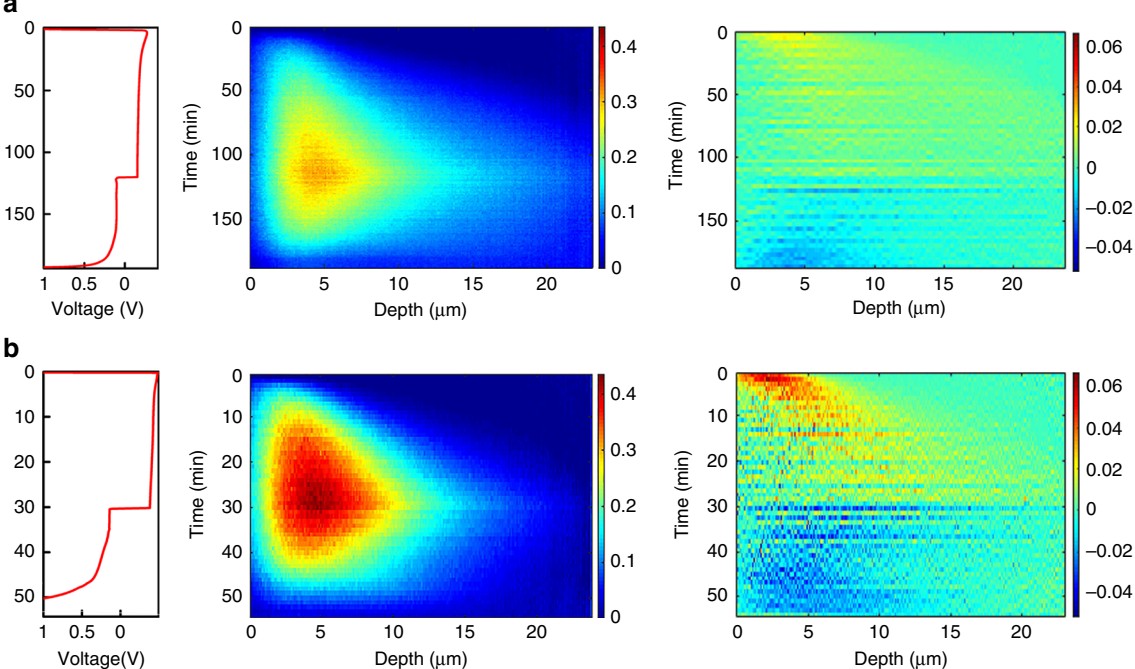

**Fig. 3** Impact of the current density on the Li density distribution. **a** Operando NDP measurements of the first plating and stripping cycle including the plating and striping activity at 0.5 mAh cm$^{-2}$ and **b** at 2 mAh cm$^{-2}$. The depth is measured starting from the interface of the Cu current collector with the electrolyte/SEI/Li-metal

will be more severe at larger overpotentials, that will influence the local properties at the current collector. Given the much less-dense dendritic growth at larger current densities, these findings suggest the existence of an optimal current, resulting in the largest plated metal density, motivating further systematic studies on the relationship between the Li-metal morphology and the current density.

As ion depletion after Sand's time, at current densities that exceed the critical current density, has been shown to initiate Li-metal dendrite growth[17–19], several strategies to prevent Li dendrite formation aim at avoiding ion depletion[11–14]. These strategies include raising the Li salt concentration in the electrolyte, inducing high transference numbers by solid electrolytes and the addition of inactive ions[11–14]. To investigate the influence of the LiPF$_6$ concentration in the EC/DMC electrolyte, the molarity was increased from 1 to 2 molar. Figure 4 shows the comparison of the Li density between 1 and 2 molar LiPF$_6$ electrolytes from NDP at different plating capacities and after stripping at 1 mA cm$^{-2}$ during the first plating/stripping cycle. Even the small increase in concentration at this relatively small current density significantly increases the plating density resulting in a thinner deposited layer. Hence, from Fig. 4 we conclude that the salt concentration may not only reduce dendrite formation[53], but also results in more compact mossy Li-metal plating. Concentrated electrolytes have been shown to reduce the thickness of the SEI layer[60], however, how the density or porosity of Li-metal films quantitatively depends on the electrolyte concentration has not been reported.

**Evolution of the total amount of Li during cycling**. Mossy growth of metallic anodes needs to be avoided to minimize the amount of electrolyte interface, resulting in electrolyte decomposition and the formation of "dead Li". Dead Li represents Li-metal that has no electric contact to the current collector, which is anticipated to remain on its original position fixed by the SEI that has formed around it. NDP allows monitoring the capacity loss

owing to SEI formation as well as "dead" Li-metal formation over repeated cycling. By integrating the Li density profiles obtained by the operando NDP, the evolution of the total amount of Li can be obtained, which allows, for instance, to monitor the amount of inactive Li after each cycle, quantified by the Li mass after stripping. To investigate the influence of current density and cycling history, two operando NDP cycling experiments were performed, shown in Fig. 5a, one starting with five cycles at 0.5 mA cm$^{-2}$ followed by five cycles at 2 mA cm$^{-2}$ and the other experiment starting with five cycles at 2.0 mA cm$^{-2}$ followed by five cycles at 0.5 mA cm$^{-2}$ all up to a 1 mAh cm$^{-2}$ plating capacity. In Fig. 5b, the integrated amount of Li in these experiments shows a marked difference for the two experiments, indicating that the initial plating current density has a significant impact on subsequent cycling.

Measuring the total amount of Li upon cycling allows to quantify the Li efficiency, here defined as the ratio between the amount of Li stripped (during oxidation of the investigated electrode) and the amount of Li plated (during reduction of the investigated electrode). The Li efficiency and the Coulombic efficiency during the two cycling experiments are shown in Fig. 5b. The Coulombic efficiency, the ratio of the integrating current during discharge and charge, does not allow to distinguish reactions that involve Li-ion transfer, including plating and stripping as well as SEI reactions, from those that do not involve Li-ion transfer, e.g., direct reduction/oxidation of the electrolyte. The Li efficiency allows to differentiate, by quantifying the amount of inactive Li, which is the sum of Li in the SEI, typically formed by one electron reduction of the EC/DMC electrolyte by the Li-metal[57,58], and "dead" Li-metal. The difference between the Coulombic efficiency and the Li efficiency quantifies the amount of irreversible reactions that do not involve Li-ion transfer, such as direct electrolyte reduction. However, it should be realized that the total amount of inactive Li quantified by NDP is a combination of "dead" Li-metal, and (in)active Li in the SEI, as the chemical nature of Li cannot be distinguished with NDP.

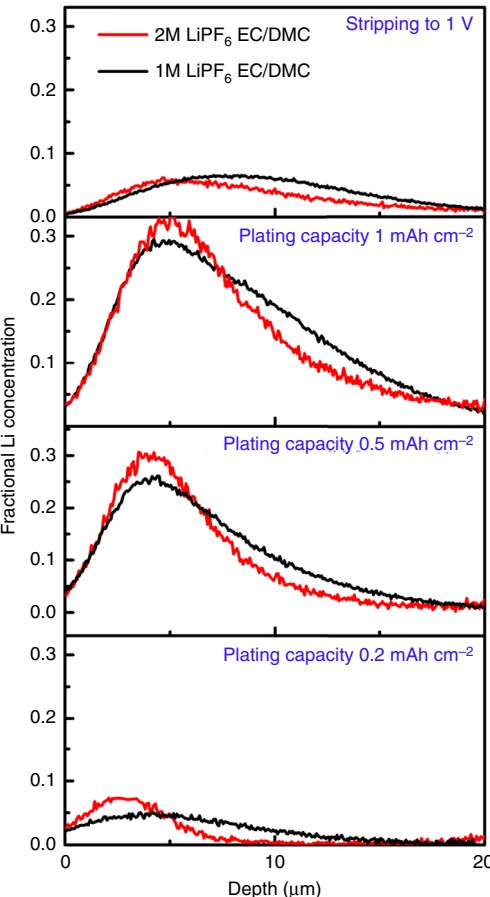

**Fig. 4** Impact of the Li-salt concentration on the Li density distribution. Li density at different stages of the first plating cycle and at the end of stripping at $1\,mA\,cm^{-2}$ comparing 1 molar and 2 molar $LiPF_6$ in EC/DMC. The depth is measured starting from the interface of the Cu current collector with the electrolyte/SEI/Li-metal

Interestingly, Fig. 5b, c demonstrate a profound history effect as the evolution of the Li density and both efficiencies depends on the sequence of applying 0.5 and $2\,mA\,cm^{-2}$. During the first cycle of both current densities, a large amount of inactive Li is observed, presumably a sum of Li in the SEI and "dead" Li-metal, becoming smaller for each subsequent cycle up to the 5th cycle. As a consequence, the initially small Li efficiencies, increase during the first five cycles for both current densities. Also the Coulombic efficiencies increase, stabilizing ~ 80%, signifying continuous SEI formation. The main difference between the first five cycles is that the Li efficiency increases to values ~ 100% for 2 $mA\,cm^{-2}$, which is significantly larger compared with the Li efficiency at $0.5\,mA\,cm^{-2}$. This indicates that the Li stripping is more efficient at a larger current density. For both current densities, the Coulombic efficiency is well below the Li efficiency, demonstrating that direct reduction of the electrolyte plays a significant role.

Comparing the two experiments, the largest changes are observed when comparing cycles 6–10. When the slow 0.5 mA $cm^{-2}$ cycling is followed by fast cycling at $2\,mA\,cm^{-2}$, again a steep increase in amount of inactive Li is observed, similar to the initial five cycles. This is consistently reflected in the drop in Li efficiency observed in Fig. 6c, which goes along with a strong decrease in Coulombic efficiency. In contrast, when fast 2 mA $cm^{-2}$ cycling is followed by slow cycling at $0.5\,mAcm^{-2}$, there is no increase in inactive Li, in fact the amount of active Li increases, which implies that a small fraction of the inactive Li is

reactivated, the origin of the larger than 100% Li efficiency, whereas the Coulombic efficiency remains constant. Clearly, slow cycling appears much more reversible after initial fast cycling as compared to initial slow cycling.

To gain more insight in the impact of current density and the cycling history on the distribution of active Li-metal and inactive Li (inactive Li-metal and Li in the SEI), the Li density profiles after each plating cycle and after each stripping cycle for both experiments are shown in Fig. 6. Comparing the densities after plating during the first five cycles at 0.5 and $2\,mA\,cm^{-2}$, Fig. 6a, c, reflects more dense plating at the larger current density, consistent with the current dependency demonstrated in Fig. 3. After five cycles at $0.5\,mA\,cm^{-2}$ a slightly larger inactive Li density is observed after stripping, see Fig. 6b, as compared with cycling at $2\,mA\,cm^{-2}$, see Fig. 6d, in particular above $10\,\mu m$ depth, consistent with the lower Li efficiency in Fig. 5c. This can be rationalized by the less-dense and more distributed Li plating at $0.5\,mA\,cm^{-2}$, which can be expected to result in more SEI formation and a larger chance on the formation of inactive Li-metal upon stripping. After the initial five cycles the evolution of the Li density after plating is profoundly different as can be observed by comparing Fig. 6a, c. Cycling at $0.5\,mA\,cm^{-2}$ during cycles 6–10 (after $2.0\,mA\,cm^{-2}$ cycling) results in more compact plating compared with initial $0.5\,mA\,cm^{-2}$ plating. Oppositely, $2.0\,mA\,cm^{-2}$ plating during cycles 6–10, after $0.5\,mA\,cm^{-2}$ results in less-dense and more distributed plating compared with initial $2\,mA\,cm^{-2}$ plating. Also after the initial five cycles the evolution of the Li density profiles after stripping is very different, demonstrating a rapid buildup of inactive Li at $0.5\,mA\,cm^{-2}$ over the full layer thickness, whereas it remains practically constant at $2\,mA\,cm^{-2}$.

These observations imply that cycling history has drastic impact on the plated Li-metal morphology upon subsequent Li-plating cycles. Based on the present observations, we propose the following mechanism, schematically shown in Fig. 6e–h. Less-dense plating at lower current densities, through less-dense nucleation[15], results in more SEI formation upon cycling as there is more electrolyte volume available. The less-dense Li-metal morphologies are more susceptible for leaving inactive Li-metal upon stripping, through the more homogeneous stripping activity shown in Fig. 3, and consistent with the lower Li efficiency and larger inactive Li density after stripping comparing Fig. 6b, d. During subsequent plating at a higher current density, the less-dense SEI/inactive Li-metal morphology formed at the low current density promotes inhomogeneous plating, and hence rapid penetration through the SEI morphology, promoting dendrite formation into the electrolyte. This will expose Li-metal to fresh electrolyte and hence initiate further SEI formation and formation of "dead" Li-metal, explaining the rapid rise of the amount of inactive Li observed in Fig. 5b. In contrast, the denser nucleation and growth of Li-metal at $2\,mA\,cm^{-2}$ is less susceptible to dead Li-metal formation and leaves less volume for SEI formation, which consequentially passivates, consistent with the stable Li and Coulombic efficiency in Fig. 5c. Perhaps the most interesting result is that this more dense morphology promotes dense plating near the current collector observed in Fig. 6c, even at lower current densities, also leading to activation of some of the inactive Li. The reactivated inactive Li may originate from reconnecting "dead" Li-metal, enabling subsequent stripping, or from reversible capacity stored in the SEI[61].

Hence, we conclude that the current dependent Li-metal plating during the initial cycles templates the SEI that forms during the first cycles. The SEI morphology formed during these initial cycles, strongly influences the Li-metal-plating morphology on subsequent cycling. These results indicate the potential

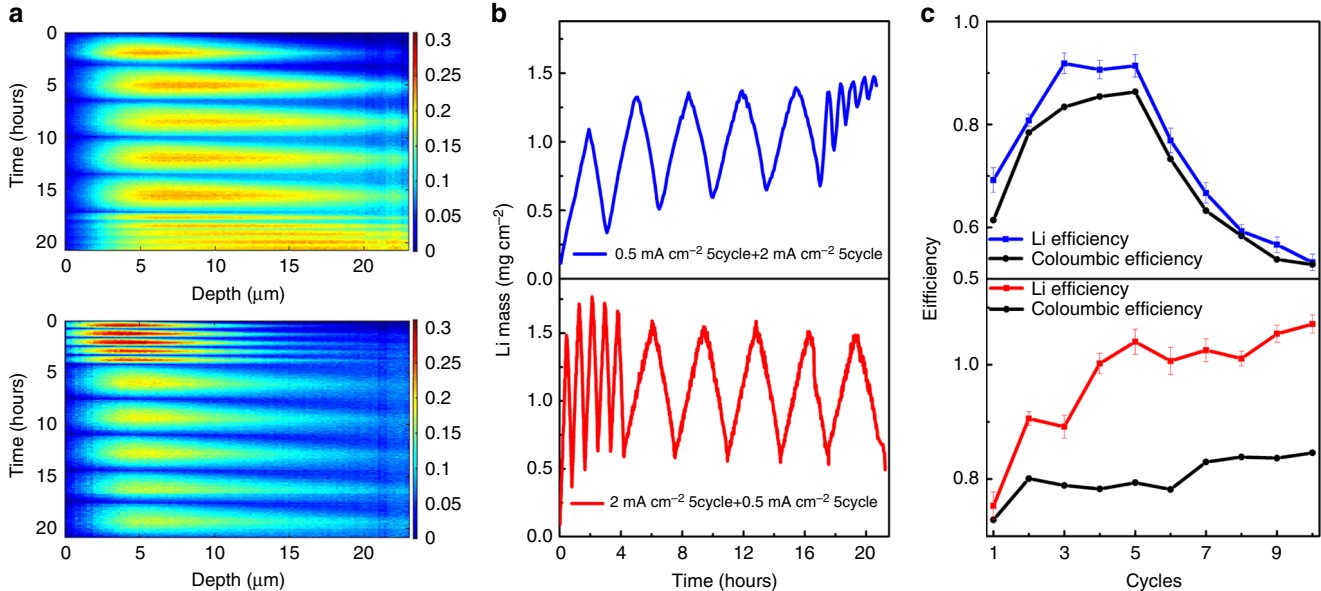

**Fig. 5** Evolution of the total amount of Li. **a** Operando NDP for five plating/stripping cycles at 2 mA cm$^{-2}$ followed by five cycles at 0.5 mA cm$^{-2}$ and for five plating/stripping cycles at 0.5 mA cm$^{-2}$ followed by two cycles at 0.5 mA cm$^{-2}$ all up to 1 mAh cm$^{-2}$ plating capacity. The depth is measured starting from the interface of the Cu current collector with the electrolyte/SEI/Li-metal. **b** Integrated amount of Li from the operando NDP experiments in **a**. **c** Coulombic efficiency and Li efficiency during the cycling

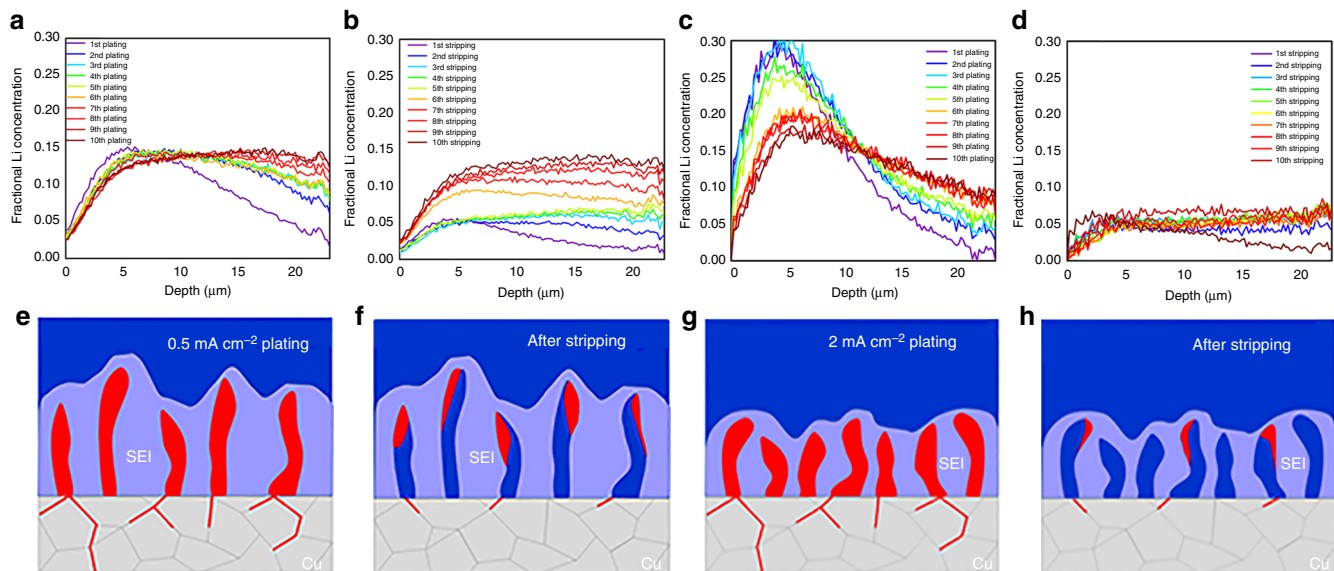

**Fig. 6** Impact of plating history on the density during plating and stripping. **a** Fractional Li density from operando NDP in Fig. 5a after plating and **b** after stripping for five cycles at 0.5 mA cm$^{-2}$ followed by five cycles at 2.0 mA cm$^{-2}$ and **c** after plating and **d** after stripping for five cycles at 2.0 mA cm$^{-2}$ followed by five cycles at 0.5 mA cm$^{-2}$. The depth is measured starting from the interface of the Cu current collector with the electrolyte/SEI/Li-metal. **e–h** Schematic representation of the plating and stripping process at 0.5 and 2.0 mA cm$^{-2}$ based on the evolution of the Li density observed with operando NDP

opportunities of initial cycling strategies to create in situ formed SEI morphologies that are more stable upon subsequent cycles.

**Evolution of the Li density in the copper current collector.** Although it is generally assumed that Li does not take up Cu, a small amount of Li uptake by Cu and other current collectors is known to occur[53,62]. Potentially, Li in the current collector can degrade both the structure and the electronic conductivity of the current collector[53,62] both of which are crucial for the functioning of Li-ion batteries. That Li actually alloys with Cu is well known from the Cu-Li phase diagram[63] from which an uptake of ~ 3.5 wt

% Li may be anticipated at 100 °C, the lowest temperature reported. However, the Li kinetics in Cu is most likely to prevent this solubility limit to be reached. Although a few studies have reported small amounts of Li-metal in Cu[53,62], the difficulty of measuring Li in current collectors and the absence of operando studies makes that current understanding of the uptake of Li in current collectors is very limited. A close look at the Cu current collector region in the operando NDP measurement shown in Fig. 1b surprisingly reveals a small amount of reversible Li uptake of the Cu current collector.

In Fig. 7a, the Cu region of the NDP measurement in Fig. 1b is shown, demonstrating Li uptake and release during four plating

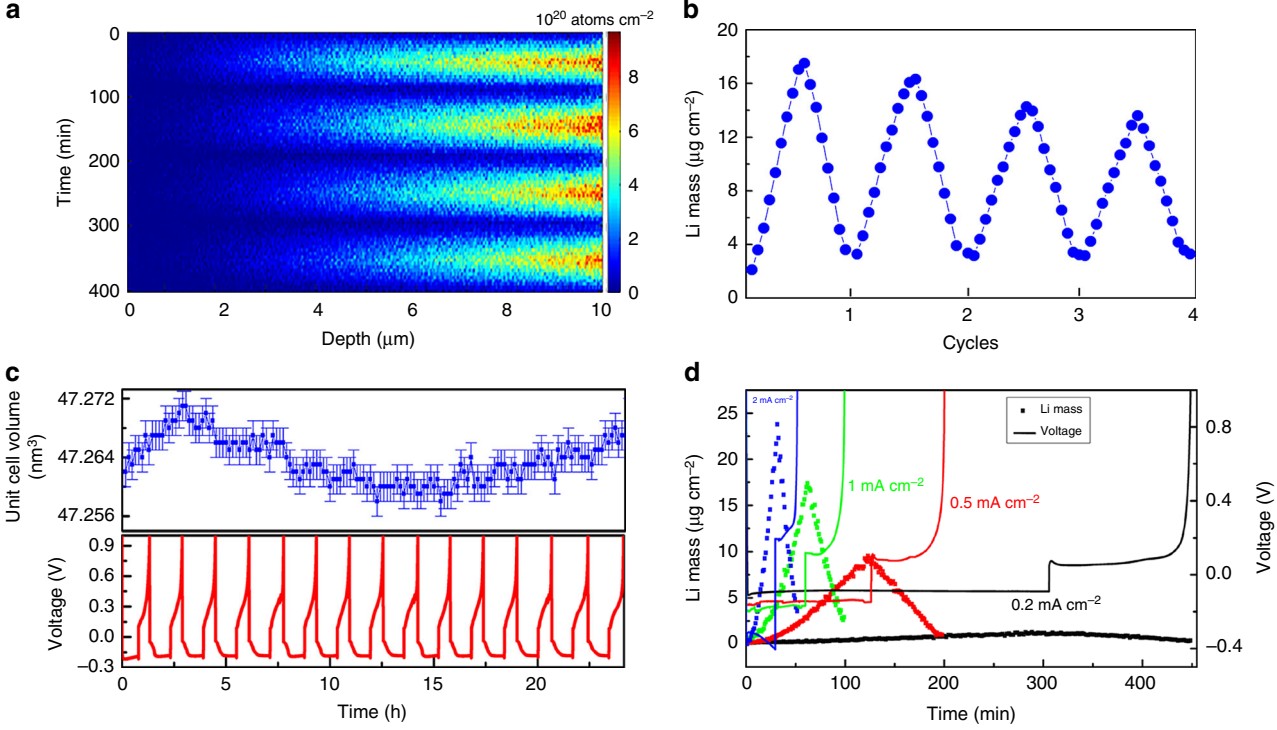

**Fig. 7** Monitoring Li in the Cu current collector during plating and stripping. **a** Operando NDP during four plating and stripping cycles, focusing on the 10 μm thick Cu current collector region from 0 to 10 μm of the measurement shown in Fig. 1b. The depth is measured starting from the interface of the of the Cu current collector with the ambient atmosphere (as indicated by Fig. 1a,b), hence the Li-metal is plated on right side in the Figure. **b** Total amount of Li per cm$^2$ in the Cu current collector during the four plating/stripping cycles. **c** Cu lattice parameter from operando XRD during 15 cycles at 1.0 mA cm$^{-2}$. The variation is the consequence of the daily temperature changes in the lab, indicating a 4 °C temperature difference during day and night. **d** Total amount of Li per cm$^2$ during the first plating/stripping cycle at different current rates, quantifying the amount of Li incorporation in Cu depending on the applied potential

and stripping cycles at 1 mA cm$^{-2}$. Li uptake in Cu was reported by post mortem analysis of the graphite anode using NDP[53]. In that case a limited penetration depth of ~ 1 μm was observed. Recently, chemical analysis showed the presence of ~ 10 μg after exposing Cu to Li-metal at 50 °C[62]. Because subsequent oxidation did not remove the Li in the Cu current collector, Li was concluded to be irreversibly trapped. The integrated amount of Li in the Cu, shown in Fig. 7b, demonstrates that a maximum of almost 20 μg cm$^{-2}$ of Li is taken up by the Cu at a deposition potential of –0.18 V. This is largely removed during stripping (oxidation) at a potential of 0.2 V, leaving 4 μg cm$^{-2}$ trapped in the Cu current collector within ~ 1 μm of the interface with the Li-metal anode. The amount and distribution of Li that is irreversible trapped appears quite similar to what was observed with post mortem analysis[64]. However, most of the Li, 16 μg cm$^{-2}$, can be reversibly added and removed from the Cu, an phenomenon that to the best of our knowledge was not reported previously. Both the irreversible and reversible amount of Li uptake by the Cu current collector strongly depends on the plating potential, as demonstrated in Fig. 7d. At 0.2 mA cm$^{-2}$, resulting in a plating potential of ~ –0.1 V, the irreversible and reversible amounts are both < 1 μg cm$^{-2}$, but at 2 mA cm$^{-2}$, resulting in a plating potential approaching –0.4 V, almost 25 μg cm$^{-2}$ is incorporated, leaving ~ 4 μg cm$^{-2}$ irreversibly trapped. Although these results clearly indicate that the plating potential drives the Li uptake of the Cu, further research is required to investigate the uptake kinetics and plating potential dependence. Interestingly, Fig. 7b indicates that subsequent cycling appears to reduce the reversible uptake of Li by Cu. At this stage it is not clear if this is due to induced changes in the grain boundaries, or SEI formation at the current collector that slows down the Li transport toward the Cu current collector. To

further investigate the nature of the Li uptake by Cu operando X-ray Diffraction (XRD) was performed during the same electrochemical plating and stripping conditions, shown in Fig. 7c. The refined Cu lattice parameter during 15 plating/stripping cycles in Fig. 7c displays a variation as a consequence of the daily temperature changes in the laboratory (which based on the thermal expansion of Cu appears to be 4 °C). However, no lattice parameter changes due to the stripping and plating potentials are observed. This indicates that Li is not taken up significantly into the Cu crystal lattice, despite the large solubility limits indicated by the phase diagram, indicating that the Li diffusion through the Cu crystal is sluggish and does not occur at room temperature. Based on this, we propose that Li is primarily transported and taken up by the grain boundary regions in the Cu current collector, as schematically shown in Fig. 6e–h. This implies diffusion of Li over several micrometers through the Cu current collector via the grain boundaries, which requires large Li-ion mobility over the surface of Cu in the grain boundaries.

**MD simulation of Li diffusion on Cu surfaces**. To gain insight in the Li mobility on the Cu surface, molecular dynamics (MD) calculations based on density functional theory (DFT) were performed for Li at the dominant surfaces of Cu, the (100) and (111) surfaces, assuming that these surfaces can represent the Cu grain boundaries. First, evaluation of the energy of Li in the bulk of Cu with DFT demonstrates that Li insertion in Cu is highly unfavorable (–2.5 to –3.0 V vs. Li/Li$^+$), whereas it is favorable at the surface (0.71 V and 0.58 V vs. Li/Li$^+$ for (111) and (100), respectively) as set out in the Supplementary Methods in more detail. For the Cu(111) surface, Li stabilizes at a vertical distance of

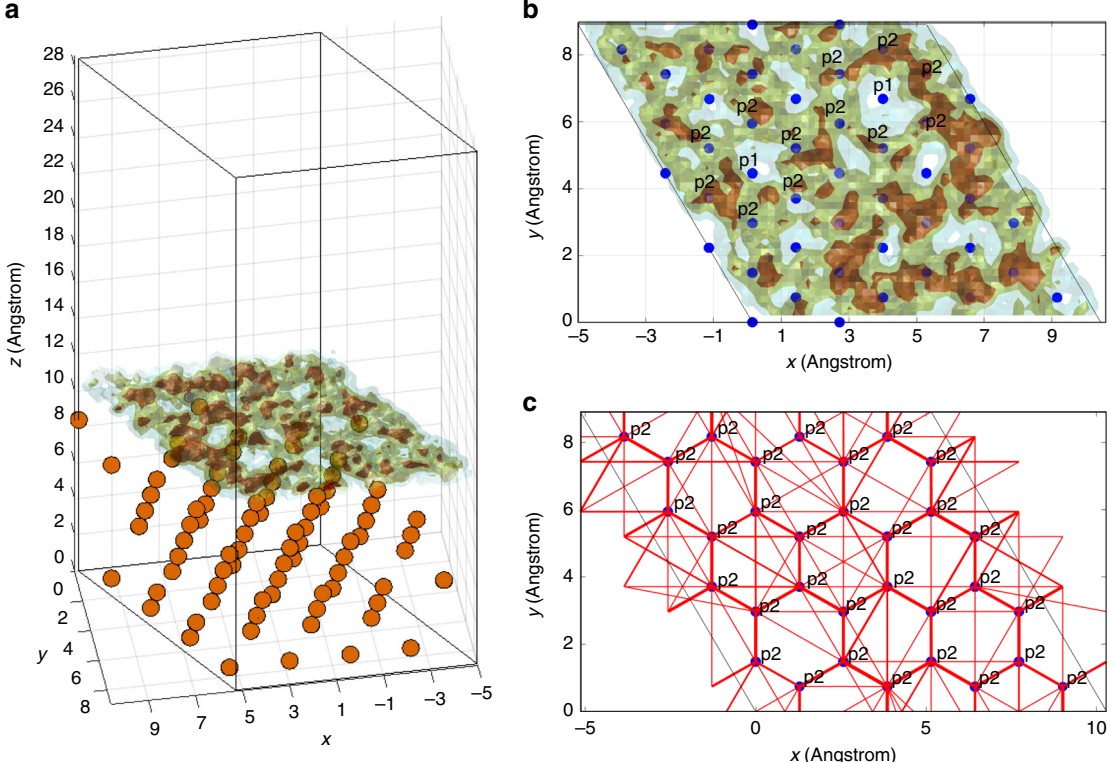

**Fig. 8** Li mobility at the Cu surface (determined by molecular dynamics Simulations). **a** Integrated Li density of 1 diffusing Li atom on the (111) copper surface resulting from a 75 ps MD simulation at 600 K. **b** Top view of the Li density indicating the Li p2 positions and the energetically less favorable p1 positions. **c** Detected transitions, represented as red lines, resulted from sampling the surface with the p2 surface positions. The thickness of the lines scales with the number of transitions

2.1 Å from the surface copper atom layer, and the most stable positions suggest that Li is diametrically projecting either Cu atoms in the second or the third copper layer, as shown in Supplementary Figure 1. Thereby the distance to the nearest three Cu atoms is maximized to 2.585 Å, which is comparable to the Cu–Cu distances (2.57 Å). For Li at the Cu(100) surface the Li position is aligned with the Cu atoms in the second Cu layer, maximizing its distance from the first copper layer as shown in Supplementary Figure 1. To evaluate the Li mobility at the Cu surfaces, MD simulations were performed. For the Cu(111) surface, a single diffusing Li diffuses over the complete 92 Å surface within 75 ps as demonstrated by the integrated Li density shown in Fig. 8a, b. In Fig. 8c, the detected transitions between the stable p2 Li positions, shown as red lines, reveal a hexagonal diffusion pathway. Even transitions via the most unfavorable p1 position occur, which is positioned 100 meV in energy above the p2 positon. This demonstrates that the energy lands scape for Li diffusion over the Cu surface is very flat, promoting high diffusivity. Similar results are obtained for more adsorbed Li-ions and for the (100) surface as shown in Supplementary Figure 2 and 4, respectively.

The activation barriers can be calculated directly from the number of transitions, given the attempt frequency[65,66]. Assuming an attempt frequency of $10^{13}\,s^{-1}$, this results in energy barriers between 26 and 49 meV between the p2 positions. Based on the mean square displacement, shown in Supplementary Figure 3, the tracer diffusion was predicted to be $4.21 \times 10^{-8}\,m^2\,s^{-1}$, indicating very rapid diffusion of Li on the surface of Cu. Details on the bulk and surface calculations can be found in the Supplementary Methods. These findings provide support that grain boundaries in the Cu current collector allow rapid Li transport, providing a rational for the observed lithiation of the Cu current collector in Fig. 7. This may give a hint that the Cu microstructure has an

important role in the Li uptake of Cu current collectors. Minimizing the presence of Cu grain boundaries at the surface of the current collector is therefore brought forward to be a potential strategy to suppress Li uptake and degradation of Cu, and perhaps also other current collectors. The present operando NDP measurement provides the possibility to study reversible and irreversible Li uptake of Cu, or of any other current collector, under the various operational conditions. This direct view on Li in current collectors can be used to develop more understanding of Li-metal uptake in current collectors and support the development of strategies that aim at preventing current collector degradation.

## Discussion

Operando observation of failure mechanisms in Li-metal anodes is crucial; however, challenged by the difficulty to measure Li under realistic working conditions in batteries. Operando NDP is demonstrated to provide quantitative measurement of the depth resolved Li density. This allows monitoring the spatial distribution during various electrochemical conditions. The present measurements give insight in the growth mechanism by the distribution of the activity, which appears to promote the formed isolated regions of inactive Li-metal as well inactive Li in the SEI during cycling. Although increasing the current density is generally expected to result in less-dense microstructures, at the relatively small currents investigated, increasing the current initially leads to more compact microstructures, rationalized by more dense nucleation induced by larger overpotentials, as schematically shown in Fig. 6e–h. The impact of the salt concentration on the Li-metal distribution is quantified, indicating that the mossy Li-metal structure is more densely plated at larger salt concentrations. This proves that the local salt concentration,

even at relatively small currents has impact on the mossy metal microstructure. The amount of inactive Li, both in the SEI and as dead Li-metal, is directly monitored by the operando NDP measurements, quantifying the larger amount of inactive Li upon increasing the current density. An interesting finding is that after relatively fast cycling, subsequent slow cycling is able to activate a fraction of the inactive Li-metal or SEI formed during the initial cycles. The results indicate that the current dependent Li-metal morphology, forming upon plating during initial cycles, templates the SEI that forms concurrently. The resulting SEI morphology has large impact upon subsequent cycling, indicating strong history effects that potentially can be used reach improved performances. By effectively measuring the Li efficiency, the ratio between the Li-metal that can be stripped during oxidation and the Li deposited during reduction, complementary information to the Coulombic efficiency is obtained. This offers new possibilities to assess the formation of the SEI and dead Li during battery operation. The electrochemical conditions of the Li-metal plating and stripping give detailed insight in unexpected reversible Li insertion in the Cu current collector. Although small uptake of Li is known, potentially aging the current collector, it is believed to be irreversibly trapped. The present operando NDP experiments show that the majority of the Li uptake in Cu is reversible, almost penetrating the 10 µm thick Cu current collector. Complementary operando XRD indicates that the uptake is not due to solubility in the Cu crystallites, but in the Cu grain boundaries, providing direction to the design of Li resistance current collectors. The present operando NDP results provide quantitative insight in Li-metal plating and stripping processes, complementary to existing microscopic and optical experiments. We believe that the direct sensitivity towards Li, the possibility to do noninvasive, non-destructive operando experiments in realistic battery operation conditions will contribute to the understanding of Li-metal anodes and the design and assessment of safe high performance future anodes.

## Methods

**Preparation of operando batteries and electrochemical tests**. A pouch cell was fabricated with ~ 10 µm thick Cu foil as the working electrode and the window towards the NDP detector. The separator uses was a 300 µm glass fiber (Whatman) sandwiched between two 25 µm PE (Celgard) sheets. In total, 500 µL conventional carbonate electrolyte (1 M LiPF6 in 1:1 v/v EC:DMC) was added to the separator sandwich. Approximately 500 µm Li-metal foil, 95% wt% [6]Li and 5% wt% [7]Li (density 0.47 g cm$^{-3}$), serves as both the counter electrode and reference electrode. Galvanostatic cycling was performed by deposition of Li onto the Cu working electrode with different current densities up to a capacity of 1 mAh cm$^{-2}$, followed by Li stripping at different current densities up to 1 V.

**NDP experiments and data handling**. NDP was performed on one of the thermal neutron beam lines at the Reactor Institute Delft. The stable isotope [6]Li can undergo a neutron capture reaction. This reaction between a neutron and the atoms' core produces two new particles; He$^{2+}$ (E$_k$ = 2044 keV) and $^3$H$^+$ (E$_k$ = 2727 keV), emitted in all directions. As these particles travel through the sample energy is lost owing to interactions with electron density. Owing to the higher mass and valence state as well as their lower initial energy the helium ions experience a larger stopping force, which prevents them from reaching the detector in the present experiments[67]. The detector is placed at 4.5 cm from the pouch cells in order to detect only the tritons ($^3$H) that are leaving the pouch cell perpendicular to the battery electrodes. As a result ~ 2% of these tritons ($^3$H) reach the detector. The energy loss of the $^3$H particles is measured with the charged particle implanted Si detector having a resolution of 3.3 keV. The energy spectrum is then collected by a Multi-Channel Analyzer. Figure 1 shows the schematic of the in situ NDP measurement setup. The NDP measurements were performed under reduced pressure (100 Torr N$_2$ equivalent), which minimizes the energy loss of the tritons through air and is enough pressure for the functioning of the pouch cells.

To relate the triton ($^3$H) energy loss and intensity to the Li depth and Li density, the data need to be corrected for the stopping power of the materials, which was determined using SRIM[67]. As the composition of the plated Li-metal electrode changes during battery cycling, the stopping power will change as a function of cycling time. In addition to Li-metal and the electrolyte an SEI will develop, each having a specific stopping power. Because the depth calibration is directly related to

the stopping power accurate knowledge of the stopping power at each state of (dis) charge (at each time frame) is required. The stopping power of the electrolyte and Li-metal is straightforwardly determined based on the well-defined density. However, for the SEI this requires detailed knowledge of the composition not generally available. To gain insight, the stopping power was calculated for a typical SEI density forming at the Li-metal anode for the presently used electrolyte (1 M LiPF6 in 1:1 EC:DMC)[68]. For the calculation of the stopping power and the depth calibration, we refer to previous work[54]. Both the stopping power of the electrolyte and the SEI as well as the resulting depth calibration are shown in Supplementary Figure 5. The depth calibration curves of the SEI and the electrolyte are very similar, a result of the very similar stopping power, both deviating at most 10% at the largest depth. Therefore, it is a reasonable assumption that the stopping power of the electrolyte and the SEI are the same, which introduces a maximal inaccuracy of 10%, but only in the most extreme case (pure SEI vs. pure electrolyte) and only at the largest depth probed by the experiments. However, the stopping power of the Li-metal is much lower compared with that of the SEI and Electrolyte. To take this into account, the data were calibrated at each time frame as follows. For each spectral point in time and energy, the ratio between the electrolyte and plated lithium is calculated by comparison with a reference spectrum of the electrolyte and a spectrum of [6]Li-enriched metal foil. Based on this the energy loss and triton flux can be directly related to the Li density as a function of depth position. It should be noted that for the small amount of Li in the Cu current collector, the stopping power is only marginally changed, and hence no significant inaccuracy is introduced for the Li depth in the Cu current collector.

It is important to realize that the neutron flux is small, ~ 10$^7$ neutrons per cm$^{22}$ per second of which 10–20% are consumed by the capture reaction in the sample. As a consequence the total amount of [6]Li consumed during the measurements is ~ 1 out of 10 billion, which is negligible and therefore NDP should be considered a nondestructive and noninvasive technique. The data are additionally corrected for the decrease in neutron flux in the sample owing to the total cross section of [6]Li[69].

**SEM**. Lithium metal-plating electrodes were prepared by discharging the punch cell to different capacity states. Before SEM imaging, the electrodes were rinsed with dimethyl carbonate in a glove box under dry Argon atmosphere and dried several times in vacuum chamber. Subsequently, samples were transferred into a SEM (JEOL JSM-6010LA) machine under dry Argon conditions, and images were taken using an accelerating voltage of 10 kV.

**Operando XRD**. In situ XRD measurements were performed using a PANalytical X'Pert Pro PW3040/60 diffractometer with Cu Kα radiation operating at 45 kV and 40 mA in an angular 2θ ranging from 30 to 95° in reflection mode geometry. Scans of 10 min were recorded on pouch cells at a current density of 1 mA cm$^{-2}$ up to a plating capacity of 1 mAh cm$^{-2}$, followed by Li stripping at different current densities up to 1 V.

**DFT calculations**. DFT calculations were performed, as implemented in the Vienna ab initio simulation package (VASP)[70,71], with projector augmented wave[72] potentials and the Perdew-Burke-Ernzerhof[73] generalized gradient approximation functional. The force convergence on each atom was set to an accuracy of 0.01 eVÅ$^{-1}$ and the energy convergence of the electron densities was set to an accuracy of 10$^{-6}$ eV. Geometry optimization and total energy calculations of the bulk copper were performed with a high cutoff energy of 600 eV and a Monkhorst-Pack[74] × 11 k-point mesh. Spin polarized calculations resulted in zero total spin moment in agreement with previous results[75], giving equivalent lattice parameters and < 2 meV difference in the obtained total copper atom energy with non-polarized calculations. Thus, non-polarized calculations were performed. Metallic smearing (ISMEAR = 1) was implemented for the partial wave function occupancies during relaxation and the "accurate" VASP setting was chosen in order to avoid wrap around errors. Subsequent self-consistent, static calculations with the Bloch-corrected tetrahedron smearing method were performed for the determination of the total energies[76]. In order to evaluate the mobility of the adsorbed Li-ions, MD simulations were performed on the Cu (111), Cu(100) supercell slab configurations. A reduced 1 × 1 × 1 k-point mesh was implemented, whereas the cutoff energy was reduced to 274 eV. Each time step accounted for 2 fs and the total simulation times were between 75 and 105 ps at 600 K. The MD simulations were performed in the NVT ensemble, with velocity scaling at every time step. Tracer diffusivities ($D^*$) were determined by measuring the mean square displacement of the diffusing Li-ion, resulting from the MD simulations.

**Data availability**. The data that support the findings of this study are available from the corresponding author upon reasonable request.

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

## Acknowledgements

The research leading to these results has received funding from the European Research Council under the European Union's Seventh Framework Programme (FP/2007–2013)/ERC Grant Agreement No. [307161] of M.W. Financial support from the Advanced Dutch Energy Materials (ADEM) program of the Dutch Ministry of Economic Affairs, Agriculture, and Innovation is gratefully acknowledged. Financial support of the KNAW for the joint research project under the scientific cooperation between China and the Netherlands, project number 530-5CDP10 is gratefully acknowledged. We are grateful to the financial support by the Natural Science Foundation of China (No.61176003).

## Author contributions

S.L. prepared the cells, and S.L. and T.V. performed the experiments and Y.X. helped with the experiments. T.V. developed the NDP analysis, and S.L. and M.W. analyzed and interpreted the NDP data. S.L. performed and analyzed the XRD experiments. Zha.L. performed the SEM experiments, F.O. provided technical support for the experimental work and A.V. performed the DFT simulations. M.W. and S.L. wrote the paper. Zhe.L. advised on the manuscript and contributed to the discussions. M.W. designed the experiment and supervised the project.

## Additional information

**Competing interests:** The authors declare no competing interests.

