## [Peer Review File · Nature Communications]

Reviewers' comments:

Reviewer #1 (Remarks to the Author):

This is a very interesting paper. Very detailed analysis. A few comments at below.

Regarding Figure 5b and paragraph starting in line 301. The authors state that "The steep increase in the amount of inactive Li after stripping, observed in Figure 5(b)... indicating progressive buildup of "dead" Li metal and Li containing SEI species." While this sentence is generally true from the data, it is vague considering the actual data presented in Figure 5. It seems like "dead" Li build up; occurs and exacerbated only when the charging/and discharging rates were increased. On the other hand, when the cell was cycled from a faster rate to a slower rate, there seems to be no "dead" Li after the first 5 fast cycles. Is there a way to distinguish disconnected Li from Li in SEI?

It is incorrect to cite [47] for NDP work while it is indeed about operando electron paramagnetic resonance (EPR) spectroscopy.

J. Wandt, C. Marino, H.A. Gasteiger, P. Jakes, R.-A. Eichel, J. Granwehr, Operando electron paramagnetic resonance spectroscopy - formation of mossy lithium on lithium anodes during charge discharge cycling, *Energy & Environmental Science* 8 (2015) 1358-1367.

It is also non-inclusive to only cite authors' own NDP work at the place NDP is firstly introduced. There are plenty of other NDP work in the literature, just name a few.

[1] Harks, P. P. R. M. L., F. M. Mulder, and P. H. L. Notten. "In situ methods for Li-ion battery research: A review of recent developments." *Journal of power sources* 288 (2015): 92-105.

[2] Profiling lithium distribution in Sn anode for lithium-ion batteries with neutrons, J Wang, DX Liu, M Canova, RG Downing, LR Cao, *Journal of Radioanalytical and Nuclear Chemistry* 301 (1), 277-284

[3] Neutron depth profiling technique for studying aging in Li-ion batteries, SC Nagpure, RG Downing, B Bhushan, SS Babu, L Cao, *Electrochimica Acta* 56 (13), 4735-4743

Line 154-156: It is better to clarify how much of 2727 keV will be lost from passing through Cu. It is an quick SRIM simulation.

Line 161-162: sentences starts "This which ...". Edit.

Why there is a discontinuity in Fig 2(j)?

Please label the depth (in fig 3,4,5,6) as depth from where or which surface/interface?

Reviewer #2 (Remarks to the Author):

Major revisions should be considered before the paper can be accepted for publication in *Nature Communications*. Detailed comments can be found below:

1. The authors present the spatial depth profiles of Li-metal anode during plating/stripping at different current densities and Li-salt concentration, in different cycles and the unclear Li-trapping in Cu current collector. Though fantastic spatial images are shown, however, the accuracy of depth is debatable. Because to get the real depth profile, one should know very well the layer composition, density, Li concentration, and so on. However, in the present work, the induced Mossy Li, dendrites and SEI are all in porous or not compact microstructures. In particular, the chemical composition of SEI is highly uncertain. These effects will potentially induce uncertainty in the converted depth profiles.

2. In the introduction, in line 35, "The positive and negative electrodes in Li-ion batteries act as host for the insertion of Li". The author should modify the description. This is because not all of electrodes

are operating by means of inserting by Li-ions, for example Si and metallic Li.

3. In Figure 5, the authors define the term "Li efficiency" and explain that dead Li can be reactivated during cycling so that the Li efficiency in Figure 5c (red line) exceeds 1. Actually, this explanation may generate some confusion since SEI kinetically forming/decomposing can also result in such effect. For instance, during Li stripping, the Li in SEI layer can, to some extent, be re-extracted since the instability of SEI. The author also denotes that the NDP is not able to distinguish the chemical difference between Li metal, "dead Li" and the Li in SEI. Therefore, a more relevant discussion should be reconsidered.

4. The authors should carefully check the units in Figure 7a, where the Li-density is indicated.

5. The authors report Li trapping in the Cu current collector (Figure 7). As shown in Figure 7a, Li is favorably trapped close to the left-hand side of Cu, while metallic Li is plated at the right-hand side. If this is the case, Li-ions should be quite moveable in the Cu current collector, a detailed discussion about Li transport through Cu is therefore required.

We would like to thank both reviewers for their important and useful comments and suggestions, which has in our view significantly raised the quality of the manuscript and will take away the concerns of the reviewers. Each suggested revision and comment, brought forward by the reviewers was accurately incorporated and considered. Below the comments of the reviewers are addressed point by point and the revisions are indicated.

Reviewer #1 (Remarks to the Author):

1. This is a very interesting paper. Very detailed analysis. A few comments at below.

Regarding Figure 5b and paragraph starting in line 301. The authors state that "The steep increase in the amount of inactive Li after stripping, observed in Figure 5(b)... indicating progressive buildup of "dead" Li metal and Li containing SEI species." While this sentence is generally true from the data, it is vague considering the actual data presented in Figure 5. It seems like "dead" Li build up; occurs and exacerbated only when the charging/and discharging rates were increased. On the other hand, when the cell was cycled from a faster rate to a slower rate, there seems to be no "dead" Li after the first 5 fast cycles. Is there a way to distinguish disconnected Li from Li in SEI?

We thank the reviewer for the positive words. As NDP does not allow to distinguish the chemical nature of the Li, we cannot disconnect Li from Li in the SEI. However, a plausible explanation for the rise in "dead" Li after increasing the cycling rate is as follows. Because of the less dense SEI morphology after slow cycling, this leaves more fresh electrolyte to be decomposed on subsequent fast cycling, which initiates more dendritic Li metal extending into the fresh electrolyte regions. This explains the rapid increase in inactive Li observed in Figure 5b. In contrast, the more dense plating at the larger current density results in a more compact Li-metal/SEI morphology. This acts as a template for subsequent slow plating, resulting in more dense plating even at low current densities as demonstrated in Figure 6. This will result in significantly less exposure of Li-metal to fresh electrolyte, explaining the stabilization of the amount of inactive Li-metal after the five initial high rate cycles observed in Figure 5b. Aiming at a more clear discussion around Figure 5b, the discussion has been revised, more clearly bringing forward this reasoning.

2. It is incorrect to cite [47] for NDP work while it is indeed about operando electron paramagnetic resonance (EPR) spectroscopy. J. Wandt, C. Marino, H.A. Gasteiger, P. Jakes, R.-A. Eichel, J. Granwehr, Operando electron paramagnetic resonance spectroscopy - formation of mossy lithium on lithium anodes during charge discharge cycling, Energy & Environmental Science 8 (2015) 1358-1367.

We apologize for this mistake, and thank the reviewer for noting this, reference [47] is corrected.

3. It is also non-inclusive to only cite authors' own NDP work at the place NDP is firstly introduced. There are plenty of other NDP work in the literature, just name a few.

[1] Harks, P. P. R. M. L., F. M. Mulder, and P. H. L. Notten. "In situ methods for Li-ion battery research: A review of recent developments." Journal of power sources 288 (2015): 92-105.

[2] Profiling lithium distribution in Sn anode for lithium-ion batteries with neutrons, J Wang, DX Liu, M

Canova, RG Downing, LR Cao, Journal of Radioanalytical and Nuclear Chemistry 301 (1), 277-284
[3] Neutron depth profiling technique for studying aging in Li-ion batteries, SC Nagpure, RG Downing, B Bhushan, SS Babu, L Cao, Electrochimica Acta 56 (13), 4735-4743

We agree, and have included the references suggested by the reviewer.

4. Line 154-156: It is better to clarify how much of 2727 keV will be lost from passing through Cu. It is an quick SRIM simulation.

We thank the reviewer for this useful suggestion. Accordingly the energy loss due to the Cu in the manuscript, which amounts 1040 keV determined from SRIM.

5. Line 161-162: sentences starts "This which ...". Edit.

The typo is corrected, and the manuscript has thoroughly been checked on other typo's.

1. Why there is a discontinuity in Fig 2(j)?

The discontinuity is due to changing from plating to stripping. The Figure shows the plating activity, so the change in Li density. During plating this is positive (yellow/red) and negative during stripping (blue). A line has been added in the manuscript to make this more clear.

2. Please label the depth (in fig 3,4,5,6) as depth from where or which surface/interface?

We thank the reviewer for this useful suggestion, and added an indicated in each figure capture from what interface the depth is measured.

Reviewer #2 (Remarks to the Author):

Major revisions should be considered before the paper can be accepted for publication in Nature Communications. Detailed comments can be found below:

1. The authors present the spatial depth profiles of Li-metal anode during plating/stripping at different current densities and Li-salt concentration, in different cycles and the unclear Li-trapping in Cu current collector. Though fantastic spatial images are shown, however, the accuracy of depth is debatable. Because to get the real depth profile, one should know very well the layer composition, density, Li concentration, and so on. However, in the present work, the induced Mossy Li, dendrites and SEI are all in porous or not compact microstructures. In particular, the chemical composition of SEI is highly uncertain. These effects will potentially induce uncertainty in the converted depth profiles.

We thank the reviewer for bringing up this very important point. Indeed the depth depends on the stopping power of the material, which is determined by the composition. From the well-known Li metal and electrolyte composition the stopping power is straightforwardly calculated. As indicated in the methods section of the manuscript, for each spectral point in time and energy, the ratio between the electrolyte and plated lithium is calculated by comparison to a reference spectrum of the electrolyte and that of the ^6Li enriched metal foil. Based on this the energy loss and triton flux can be directly related to the Li density as a function of depth position, also resulting in the Li metal porosity. However, as commented by the reviewer, the SEI will introduce an uncertainty in the depth if not taking into account the stopping power of the SEI. To estimate this uncertainty we have determined the stopping power and the depth calibration curve for a typical SEI composition (as found at the Li-metal surface for the present electrolyte) [Vatamanu et al. *The Journal of Physical Chemistry C* **2012**, 116, (1), 1114-1121]. Comparing the depth calibration for the pure electrolyte and the SEI, shown below, the difference is concluded to be relatively small.

The comparison quantifies that the maximum error in the depth will be approximately 10% at the largest depth (at smaller depth the error is smaller) and at the most extreme case if the layer is completely composed of SEI or electrolyte. In view of this we believe that the depth profiles provided are accurate, well below 1 micrometer. Note that for the Li in the Cu the error in the depth is negligible because the small fraction of Li in the Cu leads to an negligible change in the density, and therefore in the stopping power. We have added the comparison of the stopping power and depth calibration curve of the electrolyte and SEI in the supporting information, and the above results and discussion on the depth accuracy in the methods section of the manuscript.

2. In the introduction, in line 35, “The positive and negative electrodes in Li-ion batteries act as host for the insertion of Li”. The author should modify the description. This is because not all of electrodes are operating by means of inserting by Li-ions, for example Si and metallic Li.

We agree, and have modified this line in “The positive and negative electrodes in Li-ion batteries are able to store Li, the specific weight...”

3. In Figure 5, the authors define the term “Li efficiency” and explain that dead Li can be reactivated during cycling so that the Li efficiency in Figure 5c (red line) exceeds 1. Actually, this explanation may

generate some confusion since SEI kinetically forming/decomposing can also result in such effect. For instance, during Li stripping, the Li in SEI layer can, to some extent, be re-extracted since the instability of SEI. The author also denotes that the NDP is not able to distinguish the chemical difference between Li metal, “dead Li” and the Li in SEI. Therefore, a more relevant discussion should be reconsidered.

We thank the reviewer for this valuable addition. We have revised the relevant discussion, including the possibility that also Li in the SEI can to some extent contribute to the observed reactivation, including the relevant references.

4. The authors should carefully check the units in Figure 7a, where the Li-density is indicated.

We thank the reviewer for noticing this, the missing Li-density unit is added.

5. The authors report Li trapping in the Cu current collector (Figure 7). As shown in Figure 7a, Li is favorably trapped close to the left-hand side of Cu, while metallic Li is plated at the right-hand side. If this is the case, Li-ions should be quite moveable in the Cu current collector, a detailed discussion about Li transport through Cu is therefore required.

We thank the reviewer for noticing this. Actually the figure is mirrored by an unfortunate mistake, and Li is trapped at the side of the Cu where Li is plated as is reasonable to expect. The figure horizontal axis is corrected and we apologies for this confusing mistake. However even for the corrected figure, the reviewer brings forward the interesting and important point of Li kinetics: a large Li mobility is required to diffuse through several micrometers of Cu. As already discussed in the manuscript, we argue that this is most likely occurs through the grain boundaries. To further support this we have performed DFT simulations, and added these results to the revised manuscript and the Supporting Information. DFT calculations show that Li adsorbs at the Cu surface at positive voltages, whereas very negative voltages are required to store Li in the bulk of Cu metal. Moreover, molecular dynamics simulations demonstrate Li to be extremely mobile over the Cu surface as shown in the figure below (which is assumed to be a similar environment compared to the grain boundaries) supporting our hypothesis that Li can diffuse through micrometers of Cu current collectors via the grain boundaries. We have added a short discussion and one figure on these results in the manuscript. More detailed information on the calculations is provided in the supporting information.

Li mobility at the Cu surface determined by Molecular Dynamics Simulations. (a) Integrated Li density of 1 diffusing Li on the (111) copper surface during the 75 ps MD simulation at 600 K. (b) Top view of the Li-density indicating the Li p2 positions and the energetically unfavourable p2 positions. (c) Detected transitions, represented by the red lines, between the p1 and p2 surface positions, where the thickness of the lines scales with the number of transitions.

REVIEWERS' COMMENTS:

Reviewer #1 (Remarks to the Author):

I don't have other issues with the revision except for how did authors do with respect to Reviewer 1 comments 2.

Authors stated such "We apologize for this mistake, and thank the reviewer for noting this, reference [47] is corrected." As a matter of fact, [47] is still not NDP work, it is merely a swapped with an irrelevant MRI paper.

For comments 3:

Authors also stated that "We agree, and have included the references suggested by the reviewer.". Again, as a matter of fact, they did nothing with respect to this but claimed that they did. I absolutely have no intention to insist on the inclusion of the references. I'd rather to think it is an overlook.

Reviewer #2 (Remarks to the Author):

The manuscript has been modified in line with the suggestions made and can therefore now be accepted for publication.

Reviewer #1

I don't have other issues with the revision except for how did authors do with respect to Reviewer 1 comments 2.

Authors stated such "We apologize for this mistake, and thank the reviewer for noting this, reference [47] is corrected." As a matter of fact, [47] is still not NDP work, it is merely a swapped with an irrelevant MRI paper.

Reply Authors: *We are very sorry for overlooking this, apparently there were two positions where we referenced to [47]. This reference (just above Figure 1) has been replaced by the correct reference number [53]*

For comments 3:

Authors also stated that "We agree, and have included the references suggested by the reviewer.". Again, as a matter of fact, they did nothing with respect to this but claimed that they did. I absolutely have no intention to insist on the inclusion of the references. I'd rather to think it is an overlook.

Reply Authors: *We are very sorry for overlooking this, also in this case there were two positions where we referenced to NDP work. The first time NDP is introduced, last paragraph of the introduction, now references to [51-55]*

Reviewer #2 (Remarks to the Author):

The manuscript has been modified in line with the suggestions made and can therefore now be accepted for publication.

Reply Authors: *We thank the reviewer for this assessment.*